# Amorphization-induced surface electronic states modulation of cobaltous oxide nanosheets for lithium-sulfur batteries

Ruilong Li[1,2,7], Dewei Rao [3,7], Jianbin Zhou[1,7], Geng Wu[1], Guanzhong Wang[2], Zixuan Zhu[1], Xiao Han[1], Rongbo Sun[1], Hai Li [4], Chao Wang[5], Wensheng Yan [5], Xusheng Zheng[5], Peixin Cui [6], Yuen Wu [1], Gongming Wang [1✉] & Xun Hong [1✉]

Lithium-sulfur batteries show great potential to achieve high-energy-density storage, but their long-term stability is still limited due to the shuttle effect caused by the dissolution of polysulfides into electrolyte. Herein, we report a strategy of significantly improving the polysulfides adsorption capability of cobaltous oxide by amorphization-induced surface electronic states modulation. The amorphous cobaltous oxide nanosheets as the cathode additives for lithium-sulfur batteries demonstrates the rate capability and cycling stability with an initial capacity of 1248.2 mAh g$^{-1}$ at 1 C and a substantial capacity retention of 1037.3 mAh g$^{-1}$ after 500 cycles. X-ray absorption spectroscopy analysis reveal that the coordination structures and symmetry of ligand field around Co atoms of cobaltous oxide nanosheets are notably changed after amorphization. Moreover, DFT studies further indicate that amorphization-induced re-distribution of $d$ orbital makes more electrons occupy high energy level, thereby resulting in a high binding energy with polysulfides for favorable adsorption.

[1] Center of Advanced Nanocatalysis (CAN), Hefei National Laboratory for Physical Sciences at the Microscale, Department of Applied Chemistry, University of Science and Technology of China, Hefei, Anhui, China. [2] Department of Physics, University of Science and Technology of China, Hefei, Anhui, China. [3] School of Materials Science and Engineering, Jiangsu University, Zhenjiang, Jiangsu, China. [4] Key Laboratory of Flexible Electronics (KLOFE) & Institute of Advanced Materials (IAM), Jiangsu National Synergetic Innovation Center for Advanced Materials (SICAM), Nanjing Technology University, Nanjing, Jiangsu, China. [5] National Synchrotron Radiation Laboratory (NSRL), University of Science and Technology of China, Hefei, Anhui, P.R. China. [6] Key Laboratory of Soil Environment and Pollution Remediation, Institute of Soil Science, Chinese Academy of Sciences, Nanjing, P.R. China. [7] These authors contributed equally: Ruilong Li, Dewei Rao, Jianbin Zhou. ✉email: wanggm@ustc.edu.cn; hongxun@ustc.edu.cn

Despite the high theoretical energy density and capacity of Li–S batteries, their practical applications are limited by the shuttle effect, which is originated from the dissolution of lithium polysulfides ($Li_2S_x$, $x \geq 4$, LiPSs) in electrolyte[1–3]. To suppress the shuttle effect, significant efforts have been made in utilizing host materials to adsorb LiPSs via chemical interactions such as polar interactions, Lewis acid–base interactions, and catenation[4–6]. For example, metal oxides such as $TiO_2$, $Co_3O_4$, $MnO_2$, and $V_2O_5$[7–10], have been demonstrated as efficient sulfur host materials for anchoring LiPSs with the binding energy of usually 2.6–3.5 eV[11]. Compared with the above values, higher binding energy tends to be favorable to promote kinetics and suppress capacity fading in Li-S batteries[12,13], therefore, further enhancement of the LiPSs adsorption capability remains an urgent need.

The adsorption of LiPSs has closely related to the surface electronic states of the substrates[14]. Moreover, ultrathin nanosheets are considered to possess excellent structures to offer abundant binding sites with LiPSs[15]. Thus, the rational modulation of surface electronic states of ultrathin structures to enhance the binding energy with LiPSs is available but still remains a substantial challenge owing to the elusive adsorption mechanism[16–19]. Recently, amorphous materials, with a special regulation of atomic arrangement, provide the possibility for fine-tuning of electronic properties towards favorable electrochemical intermediates adsorption[20,21]. Because of the broken atomic long-range order, amorphous materials present unique surface electronic states[22], which may contribute to further enhancement of the LiPSs adsorption. Therefore, combining the ultrathin structure with the amorphous structure to boost the LiPSs adsorption capability of metal oxides is highly desirable but challenging as well due to the difficulty of size and shape control of amorphous materials[23,24].

Herein, ultrathin amorphous CoO nanosheets (a-CoO NSs) with a thickness of about 10 nm were synthesized and selected as the sulfur host. X-ray absorption spectroscopy analysis reveals that the a-CoO NSs possess abundant coordinatively unsaturated metal sites and the symmetry of ligand field around Co atoms partially convert from the octahedral ($O_h$) to the tetrahedral ($T_d$) configurations. Density functional theory (DFT) calculations reveal that the adjusted Co d orbitals in a-CoO NSs result in a significantly higher binding energy toward $Li_2S_4$ than that of the crystalline counterpart. Thus, the a-CoO NSs exhibits cycling stability with only 0.034% capacity decay per cycle for 500 cycles as the cathode additives for Li–S batteries.

## Results

### Synthesis and characterization of a-CoO NSs.
In a typical synthesis, cobalt(II) acetylacetonate ($Co(acac)_2$) and potassium bromide (KBr) were firstly well dissolved in water/ethanol mixed solution and stirring for two hours. After the solution was evaporated at 80 °C, the dried mixture was transferred into a porcelain boat and annealed at 240 °C for 80 min under air atmosphere. Finally, after washing to remove KBr and centrifugation, the a-CoO NSs were obtained. Moreover, we utilize temperature control to regulate the atomic structure in CoO nanosheets, which leads to the formation of disordered atomic structure at a low temperature (240 °C) and the crystalline counterpart (c-CoO NSs, Supplementary Fig. 1) with long-range ordered atomic structure at a higher temperature (400 °C).

In Fig. 1a, the X-ray diffraction (XRD) patterns demonstrate the crystallinity of a-CoO NSs and c-CoO NSs. The a-CoO NSs display no obvious peak, indicating the amorphous structure, while the c-CoO NSs exhibits sharp peaks indexed to the crystalline CoO (PDF#48-1719, Supplementary Fig. 2). TEM (Fig. 1b) image confirmed the nanosheet morphology with the lateral size of micrometers of the a-CoO NSs. The AFM image

and its height profile in Fig. 1c show the ultrathin 2D structure of a-CoO NSs with a thickness of about 10.4 nm. High-angle annular dark-field (HAADF) images of scanning TEM (STEM) was further recorded to investigate the atomic structure of a-CoO NSs. As displayed in Fig. 1d, a-CoO NSs exhibit randomly distributed atoms with no continuous crystal lattice stripes, indicating the lack of long-range order of atomic structure. Moreover, the weak ring in the electron diffraction (ED) pattern of a-CoO NSs (Fig. 1e) further reveals its amorphous structure. HAADF-STEM images of a-CoO NSs and corresponding energy-dispersive X-ray spectroscopy (EDS) elemental mapping images (Fig. 1f) show that Co, O, and C elements are homogeneously distributed over the entire nanosheets.

### Atomic and electronic structures of a-CoO NSs.
Raman spectrum of the a-CoO NSs (Fig. 2a) show typical vibration peaks of CoO, the peaks at 478, 516, and 680 $cm^{-1}$ were identified as $E_g$, $T_{2g}$, and $A_{1g}$ modes in $O_h$ symmetry, respectively[25], which reveals that the Co–O bonding mainly in the $CoO_6$ octahedron motif. Compared with c-CoO NSs, a-CoO NSs shows widened and slightly blue-shifted Raman peaks owing to the lack of long-range order of $CoO_6$ octahedra[26]. Moreover, the Co 2p XPS spectra (Fig. 2b) further reveal that a-CoO NSs owns the characteristic peaks of Co (II) oxidation state located at 781.1 (Co $2p_{3/2}$) and 797.1 eV ($2p_{1/2}$), respectively, and two corresponding shake-up satellite peaks, which are similar with that of c-CoO NSs[27]. O 1s XPS spectra (Supplementary Fig. 3) of both a-CoO NSs and c-CoO NSs display a strong intense peak at 531.5 eV, which can be assigned to Co–O bonding in CoO[28].

X-ray absorption spectroscopy analysis was used to verify the electronic properties as well as coordination environment of the Co atoms in a-CoO NSs. The absorption edge position on X-ray absorption near-edge structure (XANES) spectra (Fig. 2c) of both a-CoO NSs and c-CoO NSs are located close to CoO, further suggesting the valence state of Co is +2. The Fourier-transformed extended X-ray absorption fine structure (FT-EXAFS) spectra (Fig. 2d, Supplementary Fig. 4) of a-CoO NSs shows two main peaks corresponding to the Co–O and Co–Co coordination shells respectively. The fitted EXAFS structural parameters in Supplementary Table 1 show that the average coordination number of Co–O shell ($N_{Co–O}$) in both a-CoO NSs ($N_{Co–O}$ = 5.5) and c-CoO NSs ($N_{Co–O}$ = 5.8) are about 6, which indicates that the two samples mainly own $CoO_6$ octahedron units[29]. Moreover, the shortest Co–Co distance of about 3.0 Å indicates that both a-CoO NSs and c-CoO NSs own the edge-sharing connection mode between $CoO_6$ octahedra[30]. The coordination number of Co–Co shell ($N_{Co}$) of a-CoO NSs ($N_{Co}$ = 8.2) is lower than that of c-CoO NSs ($N_{Co}$ = 10.1) and CoO ($N_{Co}$ = 13.3). The reduced N values of a-CoO NSs and c-CoO NSs derived from their low dimensional (2D) structures compared with CoO[31]. Moreover, the further reduced N value of a-CoO NSs derived from the low order within an extended structure of $CoO_6$ octahedra caused by increased $CoO_6$ octahedron vacancies[32]. Thus, the a-CoO NSs possess abundant coordinatively unsaturated metal sites[33,34].

Co L-edge XANES were carried out to investigate the 3d electronic configurations of $Co^{2+}$ cations in the a-CoO NSs. The Co L-edge spectrum of the a-CoO NSs shows an obvious peak-shift (0.3 eV) toward the low energy region, compared with that of c-CoO NSs (Fig. 2e). Considering that the valence states of Co cations in both a-CoO NSs and c-CoO NSs are +2, the negative peak shift can be attributed to the different symmetries of ligand field around $Co^{2+}$ cations between a-CoO NSs and c-CoO NSs[35,36]. To examine the symmetry of ligand field around $Co^{2+}$ cations in a-CoO NSs, simulations of the Co L-edge XANES spectra were performed by using CTM4XAS (Fig. 2f, Supplementary Tables 2 and 3

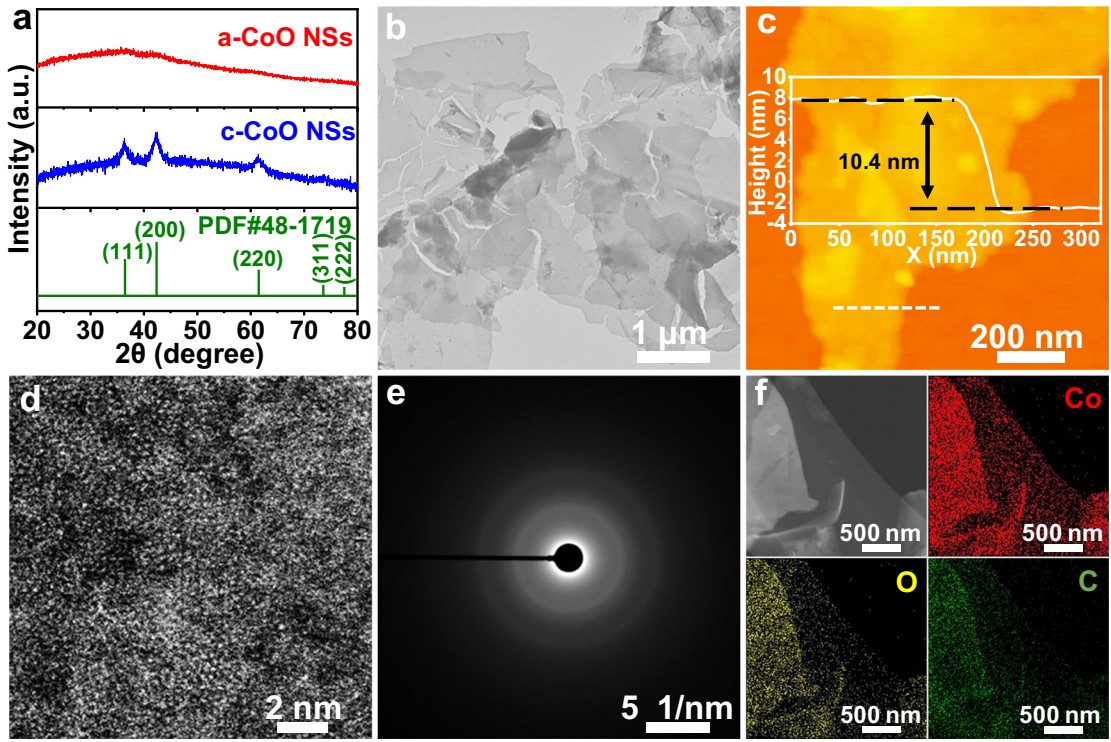

**Fig. 1 Characterization of a-CoO NSs. a** XRD patterns of a-CoO NSs and c-CoO NSs. **b** TEM, **c** AFM, and **d** HAADF-STEM images of a-CoO NSs. **e** ED pattern of a-CoO NSs. **f** HAADF-STEM image and corresponding EDS element mapping of a-CoO NSs: Co (red), O (yellow), and C (green).

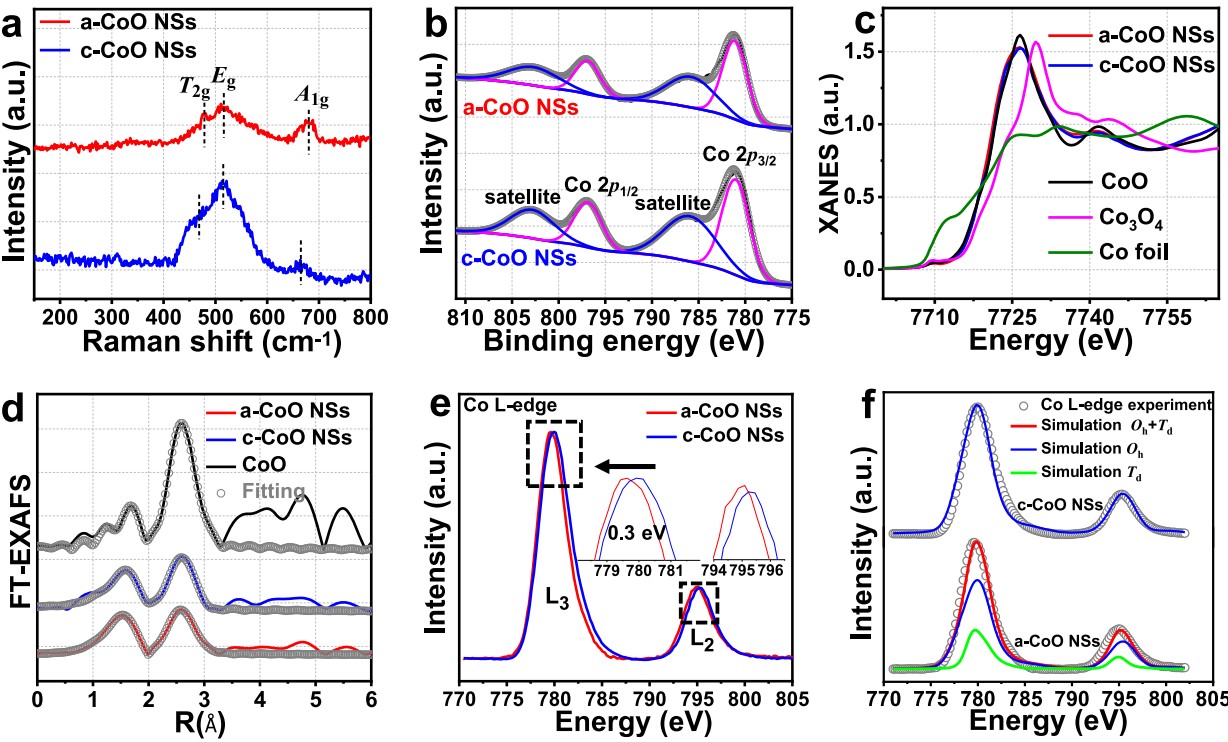

**Fig. 2 Chemical states and coordination structure analysis. a** Raman spectra of the a-CoO NSs and c-CoO NSs. **b** Co 2$p$ spectra of the a-CoO NSs and c-CoO NSs. **c** XANES spectra at the Co K edge of the a-CoO NSs, c-CoO NSs, CoO, $Co_3O_4$ samples and Co foil. **d** Fourier transformed (FT)-EXAFS spectra of the a-CoO NSs, c-CoO NSs, and CoO. **e** Co L-edge XANES spectra (inset: the amplificatory image of Co $L_3$ and $L_2$ edge respectively) and **f** simulations of Co L-edge XANES spectra for the a-CoO NSs and c-CoO NSs.

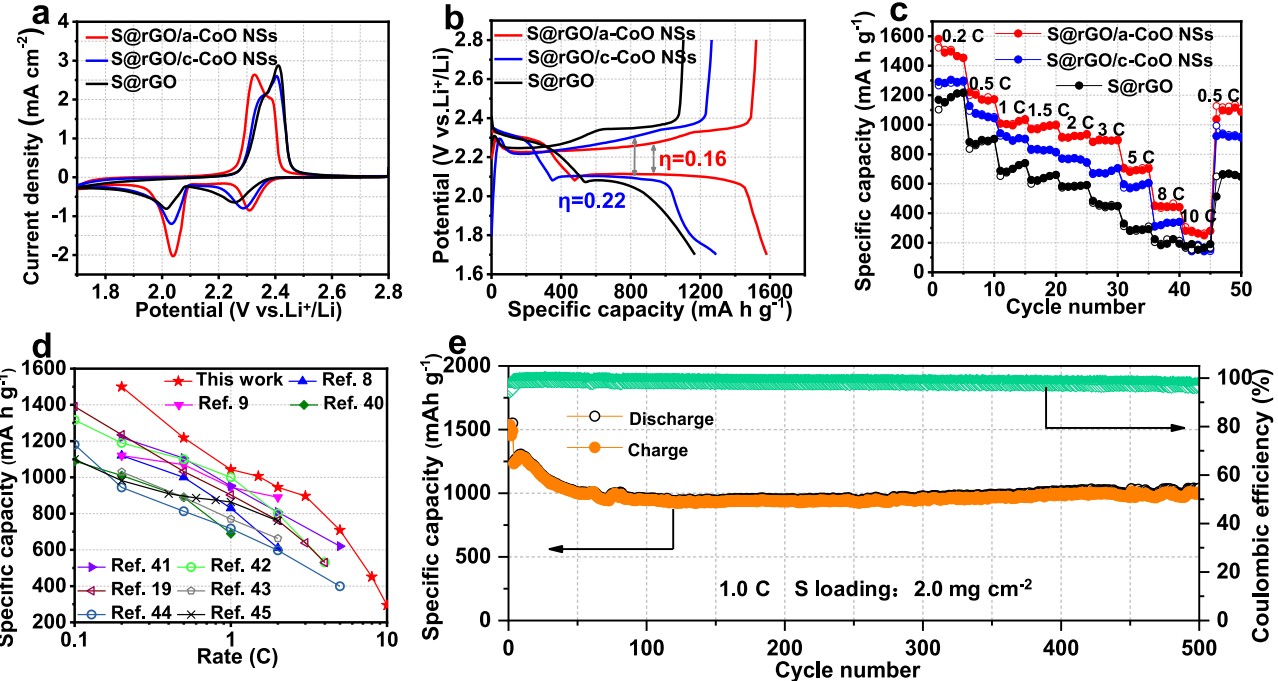

**Fig. 3 Electrochemical studies toward Li–S batteries. a** CV curves at a scan rate of 0.1 mV s⁻¹ between 1.7 and 2.8 V. **b** Galvanostatic charge–discharge profiles at 0.2 C. **c** Rate performance. **d** Comparison of the rate performance of the S@rGO/a-CoO NSs electrode in this work and other metal oxides-based electrodes recently reported. **e** Cycling performance at 1 C of the S@rGO/a-CoO NSs electrode.

and "Methods"), in which the simulations of Co $L_3$-edge spectra of Co$^{2+}$ tetrahedral ($T_d$) sites and octahedral ($O_h$) sites are located at 779.7 and 780.0 eV, respectively (Supplementary Fig. 5a, b)[37,38]. As a result, Fig. 2f shows that the a-CoO NSs possess a mixture of Co$^{2+}$ $T_d$ sites and Co$^{2+}$ $O_h$ sites with a ratio of 1: 2.3, while the c-CoO NSs only have the Co$^{2+}$ $O_h$ sites, suggesting the amorphization treatment could partially change the symmetry of the ligand field around Co atoms.

**Electrochemical performance of a-CoO NSs toward Li–S batteries.** To understand the effects of a-CoO NSs and c-CoO NSs hosts on the Li–S chemistry, electrochemical measurements were systematically carried out to compare the performance of S-loaded rGO/a-CoO NSs (S@rGO/a-CoO NSs), rGO/c-CoO NSs (S@rGO/c-CoO NSs), and S@rGO electrodes. The sulfur contents are estimated by thermogravimetric analysis (TGA, Supplementary Fig. 6). In typical CV (Fig. 3a) profiles, the positive shift in the reduction peaks and a negative shift in the oxidation peaks of the rGO/a-CoO NSs electrode indicate the improved redox kinetics compared with the rGO/c-CoO NSs electrode. It is noteworthy that the rGO/a-CoO NSs electrode shows obviously higher peak current intensity at 2.04 V than the rGO/c-CoO NSs electrode, which indicates that the conversion rate of soluble LiPSs to insoluble Li$_2$S$_2$/Li$_2$S was particularly improved[39]. The improved reduction rate of LiPSs has been demonstrated to be a crucial step in reducing the amount of LiPSs diffused in the electrolyte and improving the utilization of S and cycling life of Li-S batteries[12]. Galvanostatic charge/discharge measurements (Fig. 3b) were further performed to confirm the above results, in which the initial voltage-capacity curves of three electrodes exhibit typically two discharge and one charge plateaus, corresponding to the charge and discharge processes respectively, matching well with the CV results. A much smaller overpotential between discharge and charge plateau at the region of 2.0–2.2 V of S@rGO/a-CoO NSs electrode suggests a kinetically favorable

redox reaction of Li–S chemistry. Moreover, the charge/discharge plateau can be well maintained during cycling (Supplementary Fig. 7), reflecting the excellent capability toward LiPSs adsorption/conversion to suppress the shuttle effect. In addition, the EIS of three electrodes (Supplementary Fig. 8) has ascertained the favorable redox reactions, in which S@rGO/a-CoO NSs display the lowest charge-transfer resistance compared to the other two electrodes.

To further study their electrochemical performance at high current densities, the rate properties of the three electrodes (Fig. 3c) are measured with the rates varying from 0.2 to 10.0 C. The rGO/a-CoO NSs electrode achieves highly reversible discharge capacities of 1526.3, 1222.6, 1039.5, 938.3, and 702.3 mAh g⁻¹ at rates of 0.2, 0.5, 1, 2, and 5 C, respectively. Even at high rates of 8 and 10 C, the rGO/a-CoO NSs electrode still remains an impressive capacity of 451.5 and 306.6 mAh g⁻¹, respectively, representing a superior rate performance among the ever-reported metal oxide-based compounds for Li–S batteries (Fig. 3d, Supplementary Table 4)[7–9,19,40–45]. Moreover, at the high rate, it still possesses an obvious charge/discharge voltage plateaus (Supplementary Fig. 9), meaning the electrochemical redox kinetics have been substantially boosted with the addition of a-CoO NSs. In addition, the specific capacities of rGo, a-CoO, and c-Co are also studied, which suggests their contributions to the whole capacity are negligible (Supplementary Fig. 10). Figure 4e shows the long-term cyclability of rGO/a-CoO NSs electrode at 1 C, with an initial capacity as high as 1248.2 mA h g⁻¹ after the initial activation cycle, and retains 83.1% of the initial capacity (1037.3 mAh g⁻¹) after 500 cycles. Moreover, the discharge–charge voltage curve of the S@rGO/a-CoO NSs electrode can be well maintained after 500 cycles (Supplementary Fig. 11), further demonstrating its robustness for the chemical adsorption and conversion during cycling.

Since the electrochemical performance of Li–S battery is typically related to the adsorption properties of electrodes[14,46], we

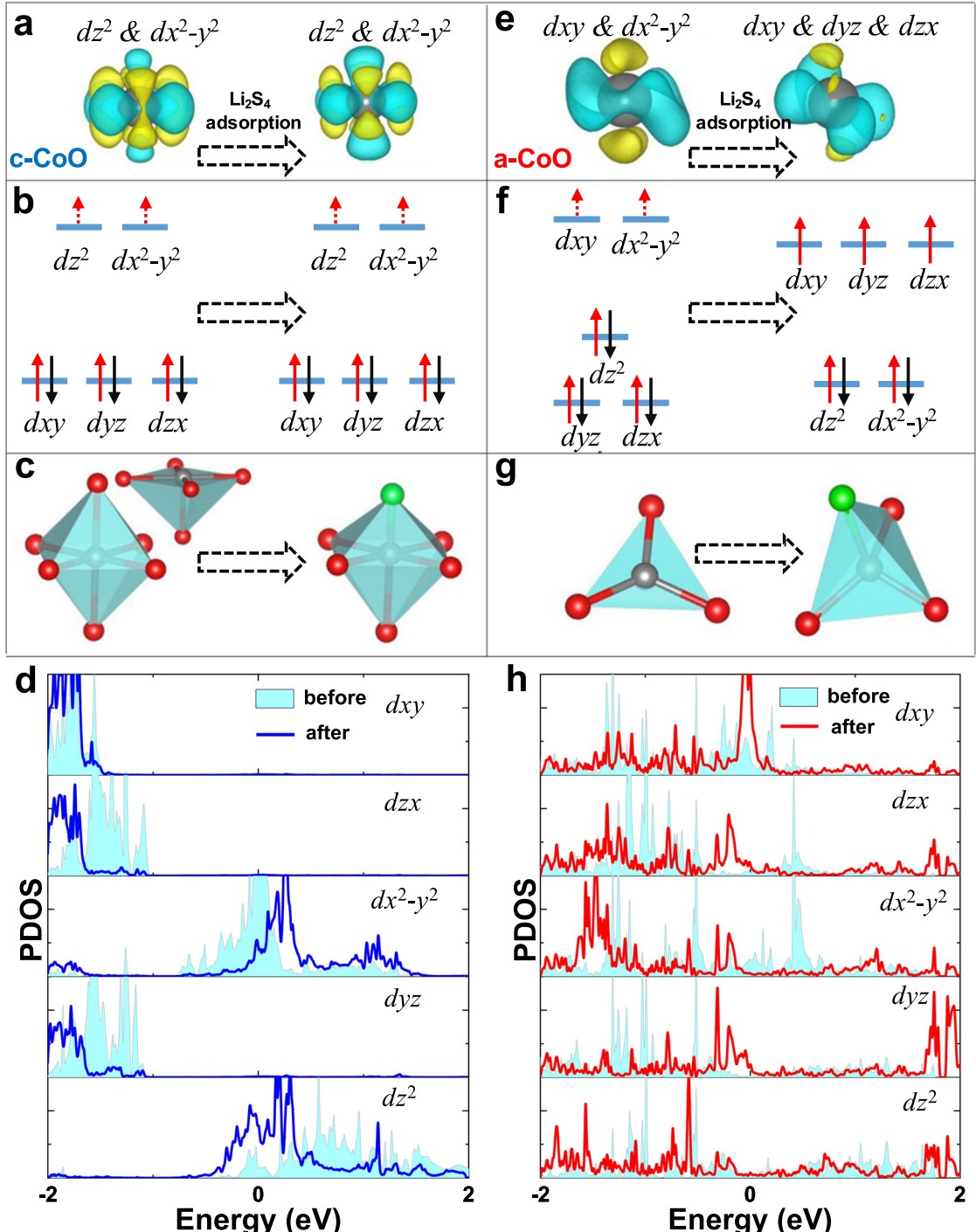

**Fig. 4 Theoretical calculations of the reaction mechanism.** The calculated deformation change density for Co of **a** c-CoO and **e** a-CoO before (left) and after (right) Li$_2$S$_4$ adsorption; the schematic of *d* orbitals for Co of **b** c-CoO and **f** a-CoO before (left) and after(right) Li$_2$S$_4$ adsorption; the coordination structure for Co of **c** c-CoO and **g** a-CoO before (left) and after (right) Li$_2$S$_4$ adsorption; the PDOS for Co of **d** c-CoO before and after Li$_2$S$_4$ adsorption respectively and **h** a-CoO before and after Li$_2$S$_4$ adsorption respectively. Red balls: O; grey balls, Co; green balls, S; yellow areas: charge accumulation; navy area: charge depletion; Fermi level in this work set to zero; the short arrow in Fig. 4b, f represents the semi-occupied state of the electron.

further compare their adsorption capabilities toward LiPSs. Therefore, HADDF-STEM was employed to map the distribution of S element in both S@rGO/a-CoO NSs and S@rGO/c-CoO NSs electrodes in the middle of discharge (Supplementary Figs. 12 and 13), which shows that the average atomic ratio of S:Co element in S@rGO/a-CoO NSs electrode is higher than that in S@rGO/c-CoO NSs electrode. Moreover, adsorption tests toward Li$_2$S$_6$ species were conducted with the same total surface area of a-CoO

NSs, c-CoO NSs, and rGO (Supplementary Fig. 14). Apparently, the solution with a-CoO NSs becomes much clearer without obvious color, suggesting the adsorption capability of CoO NSs is improved after amorphization.

**Theoretical calculations.** To understand the mechanism of the improved electrochemical performance of the a-CoO NSs, DFT

calculations were employed to investigate the properties of the c-CoO and a-CoO surfaces respectively, as well as the interaction between their surfaces and sulfide species ($Li_2S_x$, $x = 1, 2, 4, 8$). The models of the a-CoO and c-CoO surfaces are displayed in Supplementary Fig. 15a, b, where the a-CoO possesses partial distorted or truncated $CoO_4$ tetrahedrons, matching well with the simulated results of experimental Co L-edge XANES data in Fig. 2f. The changed symmetry of the ligand field around Co atoms would influence their properties of $d$ electrons. To describe the strong-correlation effects among $d$ electrons of Co, the DFT + U method with $U_{eff} = 3.3$ eV for Co was further employed. The deformation charge density of c-CoO and a-CoO (Supplementary Fig. 15c, d) show that the electron loss of Co atoms in a-CoO ($Co_{AM}$) is much different from the ones in c-CoO ($Co_{CR}$). Specifically, the electrons in high-energy orbitals are easy to loss, therefore, the $Co_{CR}$ should lose electrons on $dz^2$ and $dx^2-y^2$ orbitals, which are the highest occupied orbitals in the Co-O octahedron structure (or truncated octahedron) as in left of Fig. 4a, b. However, the electrons of $Co_{AM}$ should be lost from the high energy $dxy$ and $dx^2-y^2$ orbitals owing to the re-distributed of $d$ orbitals as in left of Fig. 4e–g. Projected density of state (PDOS) of Co in Fig. 4d, h, shows the orbital distributions and well consistent with the schematic in Fig. 4b, f, respectively. It is worth mentioning that such a difference in orbital distribution can influence the behavior of LiPSs adsorption.

To this end, we calculated the binding energy of $Li_2S_x$ on c-CoO (100) and a-CoO surfaces, respectively, as in Supplementary Fig. 16. Obviously, the $Li_2S_x$ on a-CoO surfaces have been distorted, and $Li_2S_x$ on c-CoO can maintain their geometric structures. Such difference can be reasonably explained by the strong adsorption ability of a-CoO toward $Li_2S_x$, as shown in Supplementary Table 5. The binding energies of $Li_2S_x$ on a-CoO surfaces are much higher than those on c-CoO (Supplementary Fig. 17), which is consistent to the results of the adsorption tests in Supplementary Figs. 12–14. To explore the origin of the enhanced binding energy by a-CoO, we investigated the electrons transfer between $Li_2S_4$ and a-CoO as well as c-CoO, respectively in Supplementary Fig. 18a, b. More electrons were transferred from a-CoO to $Li_2S_4$, which can strengthen the Coulomb interactions between negative charged S and positive charged Co atoms, as well as O and Li. The deformation charge density for Co atoms in $Li_2S_4$ adsorbed sites was further calculated to deeply understand the electron transfer. In the right of Fig. 4a, e, the electron loss of $Co_{CR}$ after $Li_2S_4$ adsorption is almost the same as ones without $Li_2S_4$, which can be attributed to the hardly changed orbital distributions. The PDOS in Fig. 4d indicates that only few $t_{2g}$ ($dxy$, $dxz$, $dyz$) electrons of $Co_{CR}$ lost after $Li_2S_4$ adsorption because $t_{2g}$ electrons located at a lower energy level (under Fermi level) are stable. In contrast, owing to the re-distribution of $d$ orbitals for a-CoO, the $d$ orbitals of active Co split to $t_{2g}$ ($dxy$, $dxz$, $dyz$) and $e_g$ ($dz^2$, $dx^2-y^2$) after $Li_2S_4$ adsorption (Fig. 4f). The $t_{2g}$ electrons have high energy, which can be lost easier than $e_g$. One should be noticed that there are three electrons on the higher energy level of $Co_{AM}$, and only one electron on the higher level of $Co_{CR}$. Therefore, the $Co_{AM}$ can lose more electrons to react with $Li_2S_4$. Moreover, the PDOS in Fig. 4h also demonstrated that the Co $d$ orbitals of a-CoO samples changed after S adsorption, and such change is consistent with the schematic in Fig. 4f. The PDOS in Supplementary Fig. 18c and d provide compelling evidence, that is, the more $d$ orbitals of $Co_{AM}$ are overlapped with $p$ orbital of S, which generated higher binding energy.

## Discussion

In summary, we demonstrate the strategy of amorphization-induced surface electronic states modulation in CoO nanosheets,

which can significantly improve the performance of Li–S batteries. X-ray absorption spectroscopy analysis reveals that the coordination structures of CoO nanosheets have been changed after amorphization. DFT studies further indicate that amorphization-induced redistribution of $d$ orbitals can strengthen the binding energy of LiPSs. The a-CoO NSs delivers a high-rate performance with 1100 mAh g$^{-1}$ at 1 C and retains 700 mAh g$^{-1}$ even at 5 C. More importantly, the a-CoO NSs applied in Li–S batteries have an ultralong cycle life (500 cycles) with a decay rate of only 0.034% per cycle. This work provides a novel guideline for rational modulation of surface electronic states to optimize the adsorption of LiPSs for high energy density Li–S batteries.

## Methods

**Chemicals**. Cobalt(II) acetylacetonate (Co(acac)$_2$) were purchased from Alfa Aesar. Ethanol and KBr were obtained from Shanghai Chemical Reagents, China. Anhydrous ethanol and ammonia solution (NH$_4$OH) were purchased from Sinopharm Chemical Reagent Co., Ltd. (Shanghai, China). Deionized (DI) water from Milli-Q System (Millipore, Billerica, MA) was used in all our experiments.

**Synthesis of a-CoO NSs**. In a typical procedure, 10 mg Co(acac)$_2$ was dissolved in 9 mL ethanol. After stirring to form a homogeneous solution, KBr aqueous solution (12 mg mL$^{-1}$, 1 mL) was injected into the mixed solution slowly under ultrasound at room temperature. Next, the mix solution was under vigorous stirring for 2 h at 35 °C. Then the sample was dried in an oven at 80 °C for 3 h. The sample was placed in a tube furnace and heated to 240 °C (heating rate 5 °C/min) in air for 80 min. After washing with ethanol and DI water several times, the a-CoO NSs were finally obtained and dissolved in ethanol for further characterization.

**Synthesis of c-CoO NSs**. In a typical procedure, the as-obtained a-CoO NSs were dried in a vacuum at −35 °C for overnight. Then the powder of a-CoO NSs (100 mg) was placed in a tube furnace and heated to 400 °C (heating rate 15 °C/min) for 30 min in a stream of Ar to yield c-CoO NSs.

**Synthesis of reduced graphene oxide (rGO)**. Graphene oxide (GO) was prepared by Hummer's method[47]. The rGO was synthesized using a hydrothermal method. Typically, 25 mL anhydrous ethanol was mixed with 14 mL of 5 mg mL$^{-1}$ GO solution in a flask, followed by the addition of 1.2 mL NH$_4$OH. The mixed solution was vigorously stirred for 10 h at 80 °C, before transferring to a 40 mL autoclave. The hydrothermal reaction was performed at 150 °C for 3 h.

**Characterization**. Transmission electron microscopy (TEM) images of samples were recorded by Hitachi H-7650 at an acceleration voltage of 100 kV. Aberration-corrected HAADF-STEM images were recorded with a JEOL JEM-2010 LaB$_6$ high-resolution transmission electron microscope, which operated at 200 kV. Before HADDF-STEM tests, the S@rGO/a-CoO NSs and S@rGO/c-CoO NSs electrodes were washed once with tetrahydrofuran and then ultrasonically dispersed. Atomic force microscopy (AFM) image was captured by Dimension ICON with Bruker NanoScope V controller with Scan Asyst mode. Powder XRD patterns of all samples were tested on a Rigaku Miniflex-600 operated at 40 kV voltage and 15 mA current with CuKα radiation ($\lambda = 0.15406$ nm). X-ray photoelectron spectroscopy (XPS) was collected on scanning X-ray microprobe (PHI 5000 Verasa, ULAC-PHI, Inc.) by Al Ka radiation and the C1s peak located at 284.8 eV as standard. XAFS measurements and data analysis: XAFS spectra at the Co K-edge were recorded at BL14W1 station in Shanghai Synchrotron Radiation Facility (SSRF). The electron storage ring of SSRF was performed at 3.5 GeV with a maximum of current of 250 mA. The Co K-edge XANES data were recorded in a fluorescence mode and Co foil, CoO, and Co$_3$O$_4$ were used as references. The Raman spectrum was performed on LabRamHR. The obtained N$_2$ adsorption–desorption isotherms were evaluated to give the Brunauer–Emmett–Teller (BET) specific surface area. Thermogravimetric analysis (TGA, SDT-Q600) was performed under nitrogen atmosphere by heating from RT to 500 °C at 10 °C min$^{-1}$ to measure the sulfur content in the cathode materials with the samples degassed at 30 °C for 3 h.

**L-edge XANES measurement and simulation**. The Co L-edge XANES spectra were measured in the Hefei NSRL station. The experimental condition was at room temperature. The simulations of Co L-edge XANES spectra were using CTM4XAS program and the adjusted parameters are provided in Supplementary Tables 2 and 3, where the temperature for the simulations is 300 K[48–50].

**Electrochemical measurements**. Sulfur was loaded into rGO, rGO/a-CoO NSs and rGO/c-CoO NSs by a melt-diffusion method in a glass tube with controlled weight ratios (rGO: a-CoO NSs/c-CoO NSs = 4:1). The melt-diffusion process was conducted at 155 °C for 10 h. To fabricate the working electrodes, a slurry was prepared by mixing 80 wt% active materials (S@rGO, S@rGO/a-CoO NSs, S@rGO/

c-CoO NSs), 10 wt% super P, and 10 wt% polyvinylidene fluoride, and coating them on a carbon-coated aluminum foil. The prepared electrodes were dried in a vacuum oven at 60 °C for 24 h. The electrochemical performances were finally measured with using coin-type half cells (CR 2016). The coin-type half cells were assembled in a glovebox with lithium metal foil as the anode and 1 mol L$^{-1}$ LiTFSI and DME: DOL ($v/v = 1:1$) that containing 1 wt% LiNO$_3$ as the electrolyte. The electrolyte dosage used in the half cells was ~10 μL mg$^{-1}$ of electrolyte/sulfur (E/S) ratio. The coin-cell was tested with galvanostatic cycling on a battery test system (LAND CT2001) with a voltage window of 1.7–2.8 V (vs. Li$^+$/Li) in lithium-ion battery at room temperature. Cyclic voltammetry (CV, CHI 660e) was carried out at a scan rate of 0.1 mV s$^{-1}$ from 1.7 to 2.8 V (vs. Li$^+$/Li) at room temperature. Electrochemical impedance spectroscopy (EIS) tests were conducted on a Solartron 1470E electrochemical workstation, using potentiostatic mode at open circuit potential. The sinusoidal voltage with 10 mV amplitude and the scanning frequency from 1 Hz to 100 kHz was used.

**Visualized adsorption experiment**. Li$_2$S$_6$ solution was prepared by adding a mixture of lithium sulfide and sulfur powders with a molar ratio of 1:5 into tetraglyme, followed by vigorous magnetic stirring for 24.0 h. The concentration of Li$_2$S$_6$ solution was set as 0.5 mmol L$^{-1}$. The rGOs, a-CoO NSs, and c-CoO NSs with the same total surface area (set as 0.5 m$^2$, equal to that calculated from the product of the surface area and sample weight) were added into 1.0 mL Li$_2$S$_6$ solution, respectively. These mixtures were vigorously stirred to ensure thorough adsorption and were then kept statically until all the suspended particles settled down.

**Theoretical method**. First principle calculations were performed by density function theory (DFT) code of Vienna ab initio Simulation Package (VASP)[51]. The generalized gradient approximation (GGA) with the function of Perdew–Burke–Ernzerhof[52] was used to describe the electron exchange corrections. Some parameters were set as follow: cutoff energy, 550 eV; $k$-mesh, 5 × 5 × 1; 20 Å vacuum space for all surfaces; convergence criteria, $-0.01$ eV/Å and 10$^{-5}$ eV for force and energy, respectively. The weak interactions between CoO and polysulfides have been considered through the DFT-D3 scheme[53]. To describe strong-correlation effects among $d$ electrons of Co, the Hubbard model has been employed (DFT + U)[54], and the $U_{eff}$ for Co is 3.3 eV[55]. The amorphous models were relaxed by ab initio molecular dynamic (AIMD) simulations[56] under 800 K and a time step of 1 fs lasting 5 ps. VASPKIT, an interface for VASP calculation, provides great convenience in data processing and graphic production[57].

The binding energy for polysulfide (LiPS) is calculated as Eq. (1):

$$E_b = E_{LiPS@CoO} - E_{CoO} - E_{LiPS} \qquad (1)$$

where $E_{LiPS@CoO}$, $E_{CoO}$, $E_{LiPS}$ represent the total energies of LiPS+ adsorbed samples, CoO surfaces (c-CoO or a-CoO), and LiPS, respectively. On the basis of this definition, a more negative value of $E_b$ represents a stronger binding ability as well as greater thermodynamic stability.

The deformation charge density ($\rho_d$) can describe the charge difference (or charge transfer) between atoms and compounds, and calculated as Eq. (2): $\rho_d = \rho_{scf} - \rho_{atom}$, where $\rho_{scf}$ and $\rho_{atom}$ are the charge density of the whole system and the total charge densities of each atom in the whole system, respectively. The charge density difference ($\rho_c$), described the charge redistribution caused by the adsorption of LiPSs, was calculated by Eq. (3): $\rho_c = \rho_{LiPS@CoO} - \rho_{CoO} - \rho_{LiPSs}$, where $\rho_{LiPS@CoO}$, $\rho_{CoO}$, $\rho_{LiPSs}$ represent the total charge density of LiPSs adsorbed samples, CoO surfaces (c-CoO or a-CoO) and LiPSs, respectively.

Details of amorphous models have been constructed by the following step: (1) create a pristine c-CoO (100) surface, (2) randomly delete some Co and O atoms, (3) relax these models by using AIMD simulations under 800 K, (4) optimize the relaxed structures from AIMD by using DFT calculations, (5) select the best one among these structures, and (6) calculate the electronic properties of selected models and the adsorption of polysulfides.

## Data availability

The data that support the findings of this study are available from the corresponding authors upon reasonable request.

## Code availability

The code used in this study is free availability in the website https://www.euspec.eu/code-database.

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

## Acknowledgements

This work was supported by the National Key R&D Program of China (2017YFA0700104, 2018YFA0702001, and 2017YFA0206703), National Natural Science Foundation of China (21871238, 11704365, and 21975244), Youth Innovation Promotion Association of the Chinese Academy of Science (2018494), and Fundamental Research Funds for the Central Universities (WK2060000016).

## Author contributions

X.H. and G.M.W. conceived the idea and co-wrote the paper; R.L. performed experimental work with the help of J.Z.; D.R. performed the computational work with the discussion with G.M.W.; H.L. carried out the AFM characterization; C.W., W.Y., X.Z., and P.C. performed the XAS simulations; G.W., Z.Z., X.H., and R.S. helped with the modification of the paper. G.Z.W. and Y.W. discussed the results and commented on the paper.

## Competing interests

The authors declare no competing interests.
