## [Peer Review File · Nature Communications]

Reviewer #1 (Remarks to the Author):

This manuscript reports on a strategy of significantly improving the polysulfides adsorption capability of cobaltous oxide by amorphization-induced surface electronic states modulation. The rate capability and capacity retention in S with α -CoO NS are substantially improved compared to others. The study is interesting because the amorphous structure can be better off than the crystalline. However, the evidence for supporting the correlation between α -CoO NS and the improved performance is not provided sufficiently except the DFT calculation, which only provides the reason for the improving only when the assumption is correct. The manuscript should be revised majorly with additional experiments. Details comments are following.

- 1) The study claims that the improved the polysulfides adsorption capability can improve the performance. However, there are no physical evidences except the DFT calculation, which shows only the change in the binding energy. Please provide the existence of the polysulfides by mapping S in the electrodes in the in the middle of charge/discharge.
- 2) Also, the change of the voltage plateaus at 2.0V and 2.2V in the discharge for cycles in the three electrodes should be provided for understanding the polysulfides adsorption capability because 1st plateau ($\text{S} \rightleftharpoons \text{Li}_2\text{S}_2$) mainly relates to the dissolution of the polysulfides.
- 3) Based on DFT calculations, the α -CoO NS shows possesses abundant under coordinated metals on the surface unlikely c -CoO NS. Also, EXAFS shows similar environments. However, XPS measurement, which is really sensitive to the surface, in the α -CoO NS electrode shows the quite similar Co binding environments with the c -CoO NS without metallic Co environments. I think that this should be different. Please explain the discrepancy and discuss it.
- 4) Based on DFT calculations, the binding energy of the polysulfides of the α -CoO NS is much higher than the c -CoO NS. Does this mean that the polysulfides are always on the α -CoO NS during charge/discharge? If it does, please provide the evidence of this.

Short comments

- Surface area of the the c -CoO NS, the α -CoO NS, and RGO should be provided because the surface area can affect the capability of the adsorption.
- Electrochemical activity of the c -CoO NS and, RGO the α -CoO NS by themselves should be provided.
- Voltage curves of long-term cycles and rate capability test should be provided.
 - TGA data of the the c -CoO NS, the α -CoO NS, RGO should be provided for estimating the infused S into the electrodes.

Reviewer #2 (Remarks to the Author):

The manuscript "Amorphization-Induced Surface Electronic States Modulation of Cobaltous Oxide Nanosheets for Lithium-Sulfur Batteries" by R. Li et al. presents a joint experiment-theory effect to characterize and analyze the electro-chemical performance of a novel electrode material. Several experimental techniques such as AFM, XPS, EXAFS on the K-edge, Raman spectroscopy and cyclic

voltammetry measurements are combined with Density Functional Theory to obtain a rather detailed picture of structural changes and some of the involved electronic processes and mechanism.

This being said, I cannot recommend the manuscript in its current form for publication in Nat. Comm, although the authors tangent an interesting topic (which will be highlighted later), because the manuscript stresses the significance as high capacity material with ca. 1040 mAhg⁻¹ for metal-oxide Li-S batteries, while several other metal oxides (e.g. TixOy and MnxNi(1-x)O) with similar or high capacities after retention were reported before (see, for instance review, Liu, X.; Huang, J.-Q.; Zhang, Q.; Mai, Adv. Mater. 2017, 1601759). Also, polymer based materials can reach capacities of more than 1300 mAhg⁻¹ are cheap and non-toxic while Co based materials are toxic and expensive. Where do the authors see the technological benefits for CoO based electrodes?

Besides this, the computations are not reproducible because no computational details are provided, neither are important references. Note that the material contains Co atom with partly filled d-shells. Then, the authors focus on the explanation of the underlying mechanism of the high-performance involving those d-electrons. However, standard DFT calculations cannot accurately describe strong-correlation effects among electrons. The EXAFS simulations lack details as well (references, programs, etc.). These points require clarifications.

Furthermore, the authors should establish stronger connections between their experimental and theoretical results. With the computed binding energy, for instance, it is possible to estimate the onset of the voltage curve. Also details on the construction of the amorphous system are essential. Generally, it involves generation of many random structures, (ab initio) molecular dynamic simulations to relax the structure to global minima avoiding local minima pockets and a DFT final optimization at 0K. Although the authors' approach might work as well, it is still important to test the thermodynamics (in particular stability) of the surface to obtain reasonable adsorption energies. The experimental results focus on characterising the material. They either determine structural properties (Raman, EXAFS), or charge transfer (XPS on K-edge, 2p XPS on Co), while the electronic properties (d-electron states) are only investigated by DFT calculations (which cannot be sufficiently reviewed because no computation details were provided). However, the most striking part of this manuscript is the impact of disorder (amorphization) on the electro-chemical performance over the alteration d-states. At the same time, it is also the part with the weakest evidence in the manuscript for the reasons mentioned above because direct experimental evidence is missing

I encourage the authors to strengthen their investigation in this part. Besides providing computational details that allow to reproduce the calculations and judge their quality, the mentioned symmetry reduction of the ligand-field around the Co atoms and change of d-state energies should be experimentally detectable by, e.g., X-ray absorption spectroscopy (XAS) on the L-edge of Co. The authors have access to a synchrotron facility and should consider extending their manuscript with results from this technique. In addition, detailed information on the d-orbitals can then be confirmed by reproducing the L-edge spectra with one of the freely available XAS model Hamiltonian approaches such as CTM4XAS or CRISPY.

Point-by-point Response to the Referees' Comments

We sincerely thank the referees for carefully reviewing our manuscript and their valuable comments certainly help improve the manuscript. We also appreciated the offered opportunities to revise and resubmit our manuscript. Our point-by-point responses are presented below and the changes in the revised manuscript have been highlighted for your review.

Reviewer #1 (Remarks to the Author):

This manuscript reports on a strategy of significantly improving the polysulfides adsorption capability of cobaltous oxide by amorphization-induced surface electronic states modulation. The rate capability and capacity retention in S with a-CoO NS are substantially improved compared to others. The study is interesting because the amorphous structure can be better off than the crystalline. However, the evidence for supporting the correlation between a-CoO Ns and the improved performance is not provided sufficiently except the DFT calculation, which only provides the reason for the improving only when the assumption is correct. The manuscript should be revised majorly with additional experiments. Details comments are following.

Response: We sincerely thank the reviewer for the positive comments on our work. Following the suggestions, we have provided more evidences to reveal the correlation between a-CoO Ns and the improved Li-S performance. Details are discussed below.

1) The study claims that the improved the polysulfides adsorption capability can improve the performance. However, there are no physical evidences except the DFT calculation, which shows only the change in the binding energy. Please provide the existence of the polysulfides by mapping S in the electrodes in the in the middle of charge/discharge.

Response: We appreciate the referee's valuable suggestions. Following the suggestions, we have performed ex situ HADDF-STEM to map the distribution of S element in both S@rGO/a-CoO NSs and S@rGO/c-CoO NSs electrodes in the middle of discharge, as shown in Figure R1 and Figure R2. Apparently, S element is mainly distributed in the region containing the CoO by both EDS elemental mapping and EDS line scan. The atomic ratio of S:Co element in S@rGO/a-CoO NSs electrode (0.49) is higher than that in S@rGO/c-CoO NSs electrode (0.35), which indicate that the a-CoO NSs possesses better polysulfides adsorption capability. Besides, to further compare the LiPSs adsorption capability, we also conducted the adsorption tests of a-CoO NSs, c-CoO NSs and rGO toward Li_2S_6 species (Figure R3a). To exclude the

effect of surface area toward the adsorption capability, all the samples with the same total surface area (set as 0.5 m^2 , equal to that calculated from the product of the surface area and sample weight) were added into $1.0 \text{ mL Li}_2\text{S}_6$ solution respectively, where the measured BET surface area of a-CoO NSs, c-CoO NSs and rGO is $53.4 \text{ m}^2/\text{g}$, $41.7 \text{ m}^2/\text{g}$ and $279.3 \text{ m}^2/\text{g}$, respectively (Figure R3b and R3c). Apparently, the solution with a-CoO NSs becomes much clearer without obvious color, suggesting the adsorption capability of CoO NSs is improved after amorphization. These data have been incorporated in the revised supplement information as Supplementary Figure 12, 13 and 14.

Figure R1. (a) HAADF-STEM and the corresponding element mappings images, (b) and (c) EDS line scans of S@rGO/a-CoO NSs in the middle of discharge. The average atomic ratio of S:Co element in the CoO region is 0.49.

Figure R2. (a) HAADF-STEM image and the corresponding element mappings images, (b) and (c) EDS line scans of S@rGO/c-CoO NSs in the middle of discharge. The average atomic ratio of S:Co element in the CoO region is 0.35.

Figure R3. (a) Visualized adsorption of Li_2S_6 by pristine rGO, a-CoO NSs and c-CoO NSs with the same total surface area. N_2 adsorption-desorption curves and analysis of (b) a-CoO NSs and c-CoO NSs and (c) rGO. BET surface area of a-CoO NSs, c-CoO NSs and rGO is $53.4 \text{ m}^2/\text{g}$, $41.7 \text{ m}^2/\text{g}$ and $279.3 \text{ m}^2/\text{g}$ respectively.

2) Also, the change of the voltage plateaus at 2.0V and 2.2V in the discharge for cycles in the three electrodes should be provided for understanding the polysulfides adsorption capability because 1st plateau ($\text{S} \rightarrow \text{Li}_2\text{S}_2$) mainly relates to the dissolution of the polysulfides.

Response: We thank the referee for the constructive comments. Following the suggestions, we have provided the voltage curves for cycles in Figure R4, which present the voltage profiles at 0.5 C at the 1st and 45th cycle. Clearly, the S@rGO/a-CoO NSs display smallest polarization for the discharge plateaus at the region of 2.0-2.2V, reflecting the excellent capability toward polysulfides adsorption/conversion to suppress the shuttle effect. Meanwhile, the discharge-charge voltage curve of S@rGO/a-CoO NSs electrode can be well maintained even after 500 cycles at 1 C, demonstrating its robustness for the chemical adsorption and conversion during cycling (Figure R4d). In addition, thanks to the superior adsorption capability

of a-CoO NSs, the S@rGO/a-CoO electrode also exhibits impressive rate performance (Figure R5). Even at 10C, it still possesses an obvious charge/discharge voltage plateaus, meaning the electrochemical redox kinetics has been substantially boosted with the addition of a-CoO NSs. We have incorporated these data as Supplementary Figure 7, 9 and 11, and given corresponding discussion on it in the revised manuscript.

Figure R4. Discharge–charge voltage curves before and after rate capability tests for (a) S@rGO/a-CoO NSs electrode at 0.5 C, (b) S@rGO/c-CoO NSs electrode at 0.5 C and (c) S@rGO electrode at 0.5 C. (d) Discharge-charge voltage curves of long-term cycles for S@rGO/a-CoO NSs electrode at 1 C.

Figure R5. Discharge-charge voltage curves of (a) S@rGO/a-CoO NSs, (b) S@rGO/c-CoO NSs and (c) S@rGO electrodes at various rates from 0.5 to 10.0 C.

3) Based on DFT calculations, the a-CoO NS shows abundant under coordinated metals on the surface unlikely c-CoO NS. Also, EXAFS shows similar environments. However, XPS measurement, which is really sensitive to the surface, in

the a-CoO NS electrode shows the quite similar Co binding environments with the c-CoO NS without metallic Co environments. I think that this should be different. Please explain the discrepancy and discuss it.

Response 3: We thank the referee for the valuable comments and we are pleased to clarify this issue. Yes, XPS is a surface-sensitive technique, which can reflect the valence states of the studied elements. The XPS Co 2p spectra with similar peak positions indicates both a-CoO NSs and c-CoO NSs have very similar oxidation states of $\sim+2$ (website: <https://srdata.nist.gov/xps/selectEnergyType.aspx> or reference: J. Mater. Chem. A 5, 15356-15366 (2017)). Meanwhile, according to the XAFS (Co K-edge) and Raman spectra, it was also proved that the valence states of both a-CoO NSs and c-CoO NSs were $\sim+2$. Since DFT calculations results show the presence of abundant under coordinated metals and the symmetry reduction of the ligand field around the Co atoms in a-CoO NSs, we further performed soft X-ray absorption spectroscopy on Co L-edge (Figure R6) and detailed spectroscopy analysis using the CTM4XAS program to probe the coordination environments of Co between a-CoO NSs and c-CoO NSs (a XAS model based on Hamiltonian approach, reference: Stavistki, E. & de Groot, F. M. F. The CTM4XAS program for EELS and XAS spectral shape analysis of transition metal L edges. Micron 41, 687–694 (2010)). The simulation results of experimental Co L-edge XANES show that the a-CoO NSs includes both Co^{2+} tetrahedral (T_d) sites and Co^{2+} octahedral (O_h) sites with a 1:4 ratio (Figure R6b), while the Co^{2+} cations in the c-CoO NSs only take the O_h sites (details are discussed in the revised manuscript), which is well in agreement with the DFT models. Therefore, the CoO_6 units partially convert to the CoO_4 units in the a-CoO NSs. The Raman spectra and XAFS results have also concluded that the amorphous structure possesses CoO_6 vacancies rather than O vacancies. So, the valence state of a-CoO NSs keeps almost constant and no obvious metallic Co would appear.

Figure R6. (a) Experimental Co L-edge XANES spectra (inset: the amplificatory image of Co L_3 and L_2 edge respectively) and (b) Simulations of experimental Co

L-edge XANES spectra for the a-CoO NSs and c-CoO NSs.

4) Based on DFT calculations, the binding energy of the polysulfides of the a-CoO NS is much higher than the c-CoO NS. Does this mean that the polysulfides are always on the a-CoO NS during charge/discharge? If it does, please provide the evidence of this.

Response: We thank the referee for the valuable comment. The polysulfides are not always on the a-CoO NS during charge/discharge. In our work, the DFT calculations were used to describe the adsorption properties of polysulfide molecule. Besides the adsorption, the polysulfide molecule would also undergo the process of chemical conversion and product desorption during the charge/discharge processes. For example, after the adsorption of Li_2S_8 on the a-CoO NS surfaces, Li_2S_8 undergoes reduction with the formation of Li_2S_6 intermediate, and then Li_2S_6 would undergo desorption from the a-CoO NS surfaces. The desorbed Li_2S_6 from the a-CoO NS surfaces would dissolve in the electrolyte and might be adsorbed again for further conversion to Li_2S_4 . The similar adsorption and desorption behaviors have been demonstrated in the reference: Du, Z. et al. J. Am. Chem. Soc. 141, 3977-3985 (2019).

Short comments

- Surface area of the the c-CoO NS, the a-CoO NS, and RGO should be provided because the surface area can affect the capability of the adsorption.

Response: We thank the referee for the valuable suggestion. We have provided BET surface area data in Figure R3. The BET surface area of a-CoO NSs, c-CoO NSs and rGO is 53.4 m^2/g , 41.7 m^2/g and 279.3 m^2/g , respectively. In the adsorption tests, to exclude the effect of surface area toward the adsorption capability, all the samples with the same total surface area (set as 0.5 m^2 , equal to that calculated from the product of the surface area and sample weight) were added into 1.0 mL Li_2S_6 solution respectively. We have incorporated these data in Supplementary Figure 14 in the revised manuscript.

- Electrochemical activity of the c-CoO NS and, RGO the a-CoO NS by themselves should be provided.

Response: We thank the referee for the valuable suggestion. We have provided these data in Figure R7. Clearly, their capacity contributions are quite small, in comparison with the sulfur. We have incorporated these data as Supplementary Figure 10 in the revised manuscript.

Figure R7. The electrochemical performance of (a) a-CoO NSs, (b) c-CoO NSs and (c) rGO electrodes without sulfur.

- Voltage curves of long-term cycles and rate capability test should be provided.

Response: We thank the referee for the valuable suggestion. We have provided voltage curves of long-term cycles and rate capability test in Figure R4 and R5. The S@rGO/a-CoO NSs display minimal polarization for the discharge plateaus at the region of 2.0-2.2V after long-term cycles (Figure R4), reflecting the excellent capability of polysulfides adsorption/conversion to suppress the shuttle effect. In addition, the S@rGO/a-CoO electrode also exhibits impressive rate performance (Figure R5). Even at 10C, it still possesses an obvious charge/discharge voltage plateaus, meaning the electrochemical redox kinetics has been substantially boosted with the addition of a-CoO NSs. We have incorporated these data as Supplementary Figure 7, 9 and 11 in the revised manuscript.

- TGA data of the the c-CoO NS, the a-CoO NS, RGO should be provided for estimating the infused S into the electrodes

Response: We thank the referee for the valuable suggestion. We have provided these TGA data in Figure R8. For fabricating the S@rGO/a-CoO NSs and S@rGO/c-CoO NSs electrodes, the weight ratio of rGO and CoO NSs is 4:1, the weight ratio of added sulfur and rGO+CoO NSs is also 4:1. Thus, the infused sulfur in the S@rGO/a-CoO NSs and S@rGO/c-CoO NSs electrode was calculated to be 78.0wt% and 74.4wt%, respectively. We have incorporated these data as Supplementary Figure 6 in the revised manuscript.

Figure R8. Thermogravimetric (TG) analysis of the (a) S@rGO/a-CoO NSs, (b) S@rGO/c-CoO NSs, (c) a-CoO NSs, (d) c-CoO NSs and (e) rGO.

Reviewer #2 (Remarks to the Author):

The manuscript $\acute{E}C$; Amorphization-Induced Surface Electronic States Modulation of Cobaltous Oxide Nanosheets for Lithium-Sulfur Batteries $\acute{E}D$; by R. Li et al. presents a joint experiment-theory effort to characterize and analyze the electro-chemical performance of a novel electrode material. Several experimental techniques such as AFM, XPS, EXAFS on the K-edge, Raman spectroscopy and cyclic voltammetry measurements are combined with Density Functional Theory to obtain a rather detailed picture of structural changes and some of the involved electronic processes and mechanism.

This being said, I cannot recommend the manuscript in its current form for publication in Nat. Comm, although the authors tangent an interesting topic (which will be highlighted later), because the manuscript stresses the significance as high capacity material with ca. 1040 mAhg⁻¹ for metal-oxide Li-S batteries, while several other metal oxides (e.g. TixOy and MnxNi(1-x)O) with similar or high capacities after retention were reported before (see, for instance review, Liu, X.; Huang, J.-Q.; Zhang, Q.; Mai, Adv. Mater. 2017, 1601759). Also, polymer based materials can reach capacities of more than 1300 mAhg⁻¹ are cheap and non-toxic while Co based materials are toxic and expensive. Where do the authors see the technological benefits for CoO based electrodes?

Response: First of all, we sincerely thank the reviewer for thinking our work is interesting and we are also pleased to clarify the raised issues by the referee.

Evaluating the electrochemical performance of the sulfur cathodes is a complex system that related to many parameters of assembled cells, including sulfur content,

specific capacity, rate and cycling stability. Although some of Ti_xO_y (such as TiO_2 , TiO_{2-x} , Ti_4O_7), $Mg_{0.6}Ni_{0.4}O$ and polymer based (such as *Poly* S-O-rGO and S-DIB@CNT) sulfur cathodes exhibit almost comparable or even slightly higher capacities than our work at the low current density region, the a-CoO NSs based sulfur cathodes actually display a better rate performance, indicating the significance of amorphization strategy toward improving the redox reaction kinetics of sulfur cathode. For a better comparison, we also make a table to compare the performance with the metal oxides and polymer-based sulfur cathodes, as shown in Table R1 and Figure R9. In the revised manuscript, we have added a brief discussion on the technical benefits of a-CoO NSs for Li-S batteries.

Although pushing the performance of Li-S battery to a higher level by technical advancements is important in the field of energy storage, elucidating the structure-property relation is also important under the perspective of fundamental science. Amorphous ultrathin nanostructures with abundant active sites for the LiPSs adsorption and catalytic conversion is rarely studied to decipher the benefits of disordered structures. We found that amorphization induced alteration of Co *d*-electron states in CoO nanosheets contributes to the enhancement of the LiPSs adsorption capability, which can provide valuable insights for the rational design of high-performance Li-S batteries. Therefore, we believe our work is conceptually novel to understand the Li-S chemistry in the field of Li-S batteries, besides the good electrochemical performance.

Table R1. The electrochemical performance comparison with typical metal oxides-based and polymer-based sulfur cathodes ever reported.

Samples	Sulfur content (wt%)	Rate performance mAh g ⁻¹ at (C)	Long-term cycle performance				Reference
			Current density (C)	Initial capacity (mAh g ⁻¹)	Cycle number	Reversible capacity (mAh g ⁻¹)	
a-CoO NSs	78.0	1526.3, 1222.6, 1039.5, 938.3, 702.3, 451.5 at (0.2, 0.5, 1, 2, 5, 8)	1	1248.2	500	1037.3	This work
TiO _{2-x}	45	1400, 1150, 900, 780 at (0.05, 0.2, 0.5, 1)	0.2	1098	200	889.4	Liang, Z. et. al. ACS Nano 2014 , 8 , 5249
Ti ₄ O ₇	48	1069, 850 at (0.2, 2)	2	850	500	595	Pang, Q. D. et. al. Nat. Commun. 2014 , 5 , 4759
Ti ₄ O ₇	51	1044, 527 at (0.1, 1)	0.1	1044	100	1033.6	Tao, X. Y. et. al. Nano Lett. 2014 ,

							14, 5288
Mg _{0.6} Ni _{0.4} O	<61	887, 710, 445 at (0.5, 0.7, 1)	0.1	1545	100	~1220.6	Zhang, Y. et. al. J. Mater. Chem. A , 2013 , 1 , 295
CoO/ HCN	71.3	1220, 1106, 952, 808, 620 at (0.2, 0.5, 1, 2, 5)	0.2	1242	200	996	Wu, S. et al. J. Mater. Chem. A 5 , 17352-359 (2017)
TiO ₂ -MX ene	75	1030, 890, 770, 663 at (0.2, 0.5, 1, 2)	0.2	~904	200	~521.5	Jiao, L. et al. Adv. Energy Mater. 2019 , 1900219
Fe ₃ O ₄ -N C@ACC	70	1316, 1190, 1100, 1000, 800, 531 at (0.1, 0.2, 0.5, 1, 2, 4)	0.2	~1114	1000	780	Lu, K. et al. Adv. Funct. Mater. 29 , 1807309 (2018)
(polymer based material) Poly S-O-rG O	72	1205, 998, 927, 887 at (0.1, 0.2, 0.5, 1)	0.5	1265	500	1033.5	Park, J. et. al. Adv. Energy Mater. 2017 , 7 , 1700074
S-DIB@ CNT	63.5	1300, 1130, 990, 890, 700 at (0.1, 0.2, 0.5, 1, 2)	0.1/1	~1300/898	100 initial cycles at 0.1 C	880	Hu, G. et. al. Adv. Mater. 2017 , 29 , 1603835
poly(S- r - DIB)	/	~1180, 1073, 1046, 862, 429 at (0.1, 0.2, 0.5, 1, 2)	0.1	1225	500	635	Pyun, J. et. al. ACS Macro Lett. 2014 , 3 , 229-232
CP(S - PMAT)/ C	80.89	1240, 1085, 976, 880, 780, 600 at (0.1, 0.2, 0.5, 1, 2, 5)	0.2	1085	100	1074	Zeng, S. et. al. ChemSusChem 2017 , 10 , 3378-86
PPy-Mn O ₂	70	~1500, 1138, 868, 660, 500, 350 at (0.1, 0.2, 0.5, 1, 2, 4)	0.2	1420	200	985	Zhang, J. et. al. Nano Lett. 2016 , 16 , 7276-7281

Figure R9. The rate performance comparison with typical (a) metal oxides-based and (b) polymer based sulfur cathodes ever reported.

Besides this, the computations are not reproducible because no computational details are provided, neither are important references. Note that the material contains Co atom with partly filled *d*-shells. Then, the authors focus on the explanation of the underlying mechanism of the high-performance involving those *d*-electrons. However, standard DFT calculations cannot accurately describe strong-correlation effects among electrons. The EXAFS simulations lack details as well (references, programs, etc.). These points require clarifications.

Response: We thank the referee for the valuable suggestions. Following the suggestions, the detailed computational methods and parameters have been added in “Method” section, as well as some important references.

We agree with the referee’s comment that standard DFT calculation cannot accurately describe strong-correlation effects among *d* electrons. To address this issue, the Hubbard models are further employed to describe such effects, that is, DFT+U method. Based on the same configuration of amorphous CoO in the previous DFT calculations (Figure R10b and Figure R11), DFT+U are employed to improve the quality of calculated results. The improved data are incorporated in the revised manuscript as Fig. 4, Supplementary Fig. 15-18, and Supplementary Table 4. The numerical values for binding energy between a-CoO and Li₂S₄ obtained by DFT+U (7.58 eV) is more accurate than the result obtained by standard DFT calculations (6.44 eV). Moreover, the overall adsorption trend is consistent and the conclusions are the same. We have summarized the adsorption values toward the various sulfide species in Figure R12. In order to retain the more accurate value calculated by DFT+U, we have updated the data in the revised supporting information as Supplementary Figure 17 and Table 4. Corresponding discussions were added in the revised manuscript.

The details of EXAFS simulations (references, programs, etc.) have been provided in the notes of revised Supplementary Table 1. The EXAFS data were analyzed using the software packages Demeter (Ravel, B., Newville, M., 2005. Journal of Synchrotron Radiation 12, 537-541). The spectra were normalized using Athena

firstly, and then shell fittings were performed with Artemis. The $\chi(k)$ function was Fourier transformed (FT) using k^3 weighting, and all fittings were done in R-space. The coordination parameters were obtained by fitting the experimental peaks with theoretical amplitude.

Figure R10. The optimized structures of (a) c-CoO, (b) a-CoO; the deformation charge density of (c) c-CoO and (d) a-CoO. Red balls: O; grey balls, Co; yellow areas, charge accumulation; navy area, charge depletion.

Figure R11. The calculated deformation change density for Co of (a) c-CoO and (e) a-CoO before (left) and after (right) Li_2S_4 adsorption; the schematic of d orbitals for Co of (b) c-CoO and (f) a-CoO before (left) and after (right) Li_2S_4 adsorption; the coordination structure for Co of (c) c-CoO and (g) a-CoO before (left) and after (right) Li_2S_4 adsorption; the PDOS for Co of (d) c-CoO before (navy area) and after (blue line) Li_2S_4 adsorption and (h) a-CoO before (navy area) and after (red line) Li_2S_4 adsorption. Red balls: O; grey balls, Co; green balls, S; yellow areas: charge accumulation; navy area: charge depletion; Fermi level in this work set to zero; the short arrow in Fig. 4b and 4f represents the semi-occupied state of the electron.

Figure R12. The binding energies of Li_2S_x on a-CoO and c-CoO in the present work calculated by standard DFT and DFT+U methods, respectively.

Furthermore, the authors should establish stronger connections between their experimental and theoretical results. With the computed binding energy, for instance, it is possible to estimate the onset of the voltage curve. Also details on the construction of the amorphous system are essential. Generally, it involves generation of many random structures, (ab initio) molecular dynamic simulations to relax the structure to global minima avoiding local minima pockets and a DFT final optimization at 0K. Although the authors' approach might work as well, it is still important to test the thermodynamics (in particular stability) of the surface to obtain reasonable adsorption energies.

Response: We thank the referee for the valuable suggestions. To better describe the relation between experimental results and theoretical calculation, we have carefully analyzed these results and make the corresponding correlation. Firstly, the theoretical model is based on the experimental data of EXAFS and fitting parameters, and then demonstrated to be consistent with the results of XAS on Co L-edge (revised Fig. 2f), detail has been described in “Method” section. Secondly, more correlation discussions have been added. As in section of “Discussion”, we have expressed that: “The models of the a-CoO and c-CoO surfaces are displayed in Supplementary Fig. 15 a-b, where the a-CoO possesses partial distorted or truncated CoO₄ tetrahedrons, matching well with the simulated results of experimental Co L-edge XANES data in Figure 2f. The changed symmetry of ligand field around Co atoms would influence their properties of *d* electrons. To describe the strong-correlation effects among *d* electrons of Co, DFT+U method with $U_{\text{eff}} = 3.3$ eV for Co was further employed.”, and “The binding energies of Li₂S_x on a-CoO surfaces are much higher than those on c-CoO (Supplementary Fig. 17), which is consistent to the results of adsorption tests in Supplementary Fig. 12-14, where the a-CoO NSs possesses stronger LiPSs adsorption capability”.

In addition, it is difficult to estimate the onset of the voltage curve. Firstly, the DFT calculations (including DFT+U) study the materials properties under 0 K, and the influence of temperature is not described, but the onset voltage is typically dependent on operation temperature for Li-S battery. Secondly, the onset of the voltage is related to the reaction energies of a large amount polysulfides molecules that adsorbed on cathode surfaces. However, the DFT calculations (including DFT+U) only describe the adsorption properties of single polysulfide molecule.

We totally agree with the referee's suggestion about the method for constructing amorphous system. In our work, we have used similar method to construct amorphous model and tested its thermodynamics stability. Based on the referee's valuable advices, details of method have been described in “Method” section. Details of amorphous model have been constructed by the following step: (1) create a pristine c-CoO (100) surface, (2) randomly delete some Co and O atoms, (3) relax these models by using *ab initio* molecular dynamic (AIMD) simulations under 800 K, (4) optimize the relaxed structures from AIMD by using DFT calculations, (5) select the best one

among these structures, (6) calculate the electronic properties of the selected model and the adsorption of polysulfides. The amorphous CoO configuration is thermodynamically stable after the above-mentioned steps.

The experimental results focus on characterising the material. They either determine structural properties (Raman, EXAFS), or charge transfer (XPS on K-edge, 2p XPS on Co), while the electronic properties (d-electron states) are only investigated by DFT calculations (which cannot be sufficiently reviewed because no computation details were provided). However, the most striking part of this manuscript is the impact of disorder (amorphization) on the electro-chemical performance over the alteration of d-states. At the same time, it is also the part with the weakest evidence in the manuscript for the reasons mentioned above because direct experimental evidence is missing.

I encourage the authors to strengthen their investigation in this part. Besides providing computational details that allow to reproduce the calculations and judge their quality, the mentioned symmetry reduction of the ligand-field around the Co atoms and change of d-state energies should be experimentally detectable by, e.g., X-ray absorption spectroscopy (XAS) on the L-edge of Co. The authors have access to a synchrotron facility and should consider extending their manuscript with results from this technique. In addition, detailed information on the d-orbitals can then be confirmed by reproducing the L-edge spectra with one of the freely available XAS model Hamiltonian approaches such as CTM4XAS or CRISPY.

Response: We thank the referee for the valuable suggestion to measure the changed symmetries of the ligand field around the Co atoms. Following the suggestions, we have used CTM4XAS to reproduce the Co L-edge XAS spectra and obtained more detailed information on the Co *d*-orbitals. DFT calculations show the symmetry reduction of the ligand-field around the Co atoms in a-CoO NSs. The changed symmetry of the ligand field around the Co atoms was detected by Co L-edge XANES spectra. The Co L-edge spectrum of the a-CoO NSs shows an obvious peak-shift (0.3 eV) toward the low energy region, compared with that of c-CoO NSs (Figure R13a). Considering that the valence states of Co cations in both a-CoO NSs and c-CoO NSs are +2, the negative peak-shift can be attributed to the different symmetry of the ligand field around Co^{2+} cations between a-CoO NSs and c-CoO NSs. Thus, simulations of the Co L-edge XANES spectra were also performed (Figure R13b, Table R2 and “Methods”), in which the simulations of Co L_3 -edge spectra of Co^{2+} tetrahedral (T_d) sites and octahedral (O_h) sites are located at 779.4 eV and 780.0 eV respectively (Supplementary Fig. 5-b, references: Mu, C. et al. Adv. Mater. 32, e1907168 (2020); Stavitski, E. & de Groot, F. M. F. Micron 41, 687–694 (2010)). As a result, Fig. 2f shows that the a-CoO NSs possess a mixture of $\text{Co}^{2+} T_d$ sites and $\text{Co}^{2+} O_h$ sites with a ratio of 1:4, while the c-CoO NSs only have the $\text{Co}^{2+} O_h$ sites, suggesting the amorphization treatment could partially change the symmetry of the ligand field around Co atoms. We have added the updated results in the revised manuscript and given corresponding discussion on it.

Figure R13. (a) Experimental Co L-edge XANES spectra (inset: the amplificatory image of Co L₃ and L₂ edge respectively) and (b) Simulations of experimental Co L-edge XANES spectra for the a-CoO NSs and c-CoO NSs.

Figure R14. Co L-edge XANES spectrum of the (s) simulation of Co²⁺ T_d sites and (b) simulation of Co²⁺ O_h sites. The adjusted parameters of the simulation by CTM4XAS are given in Supplementary Table 2.

Site symmetry	Crystal field strength (10Dq)	Charge transfer energy (Δ)	Hopping e _g electrons (t_e)	Hopping t _{2g} electrons (t_t)	Hubbard U value (U_{dd})	Core hole potential (U_{pd})
Co ²⁺ O _h	1.0	1.0	2.0	1.0	6.0	5.0
Co ²⁺ T _d	-0.5	6.0	1.0	2.0	5.0	6.5

Table R2. The adjusted parameters for the simulation of Co L-edge XANES spectra by CTM4XAS.

Reviewer #1 (Remarks to the Author):

The revised manuscript is majorly updated as requested by the reviewers. I would like to recommend the publication of this manuscript.

Reviewer #2 (Remarks to the Author):

In the revision of the manuscript 'Amorphization-Induced Surface Electronic States Modulation of Cobaltous Oxide Nanosheets for Lithium-Sulfur Batteries' by Li et al., the authors have seriously improved the content of the manuscript and strengthened their analysis. Now providing computational details for the ab initio simulations and the EXAFS fitting parameters, the results are reproducible. Furthermore, the extension of the manuscript with XAS results (both experimental and theoretical) provides further evidence for the proposed reaction mechanism. The updated PDOS with the Hubbard U term applied to Co's d-states in Figure 4 provides a much clearer picture than the previous version and is additionally in agreement with the experimental and theoretical XAS analysis clarifying the role of symmetry, crystallinity and amorphization for the system.

Overall, the presented manuscript carries a detailed presentation of the amorphization-induced performance increase for Co containing nanosheets with exhaustive support from various experimental and theoretical investigations. In combination with the relatively high-performance compared to other Co-based materials, an acceptance for publication in Nature Comm. can be justified in my opinion. However, before acceptance a few points remain to be clarified in the updated manuscript:

Technical comments:

There are several cases for which the 2p XAS signal of Co²⁺ systems shows temperature dependence.^{1–3} The temperature dependence is given by low-lying multiplet initial states. Hence, it is essential to provide the temperature for the simulation, which simultaneously needs to be consistent with the experimental conditions to achieve consistency.

Moreover, the choice of Upd being smaller than Udd in the case of Co²⁺ Oh is somewhat unusual, and perhaps even unphysical. Upd is generally larger than Udd⁴ because the presence of the core hole in the final state leads to a contraction of the states resulting in stronger Coulomb repulsion (i.e. the monopole term in the Slater Condon integrals) which is usually given by a value being 1-2 eV larger than Udd. Hence, the authors should explain the choice of Upd < Udd.

In addition, Udd in the multiplet calculations and the Udd in the first principles part of the manuscript differ almost by a factor of 2. This should also be explained or addressed by the authors.

Minor comments

- The phrase 'all calculations' in the 'Theoretical method' part does not capture the complexity the authors put into this manuscript as DFT calculations are only a part of it.
- There are a few typos in the 'Theoretical method', e.g., 'a pro-', 'Hubbard models'.
- What do the authors mean by 'From the PDOS in Fig. 4d, only few t_{2g} electrons under Fermi

level lost after Li₂S₄ adsorption.?’

- Unit are missing in Table 2 in SI, as well as details on the applied convolution of transition intensities and screening parameters.

References

- (1) Csiszar, S. I.; Haverkort, M. W.; Hu, Z.; Tanaka, A.; Hsieh, H. H.; Lin, H.-J.; Chen, C. T.; Hibma, T.; Tjeng, L. H. Controlling Orbital Moment and Spin Orientation in CoO Layers by Strain. *Phys. Rev. Lett.* 2005, 95 (18), 187205. <https://doi.org/10.1103/PhysRevLett.95.187205>.
- (2) Tanaka, A.; Jo, T. Temperature Dependence of 2 p -Core X-Ray Absorption Spectra in 3 d Transition-Metal Compounds. *J. Phys. Soc. Japan* 1992, 61 (6), 2040–2047. <https://doi.org/10.1143/JPSJ.61.2040>.
- (3) de Groot, F. M. F. X-Ray Absorption and Dichroism of Transition Metals and Their Compounds. *J. Electron Spectros. Relat. Phenomena* 1994, 67 (4), 529–622. [https://doi.org/10.1016/0368-2048\(93\)02041-J](https://doi.org/10.1016/0368-2048(93)02041-J).
- (4) Groot, F. De. Multiplet Effects in X-Ray Spectroscopy. *Coord. Chem. Rev.* 2005, 249 (1–2), 31–63. <https://doi.org/10.1016/j.ccr.2004.03.018>.

Point-by-point Response to the Referees' Comments

Reviewer #1 (Remarks to the Author):

The revised manuscript is majorly updated as requested by the reviewers. I would like to recommend the publication of this manuscript.

Response: We thank the reviewer very much for the positive comments on our work.

Reviewer #2 (Remarks to the Author):

In the revision of the manuscript 'Amorphization-Induced Surface Electronic States Modulation of Cobaltous Oxide Nanosheets for Lithium-Sulfur Batteries' by Li et al., the authors have seriously improved the content of the manuscript and strengthened their analysis. Now providing computational details for the ab initio simulations and the EXAFS fitting parameters, the results are reproducible. Furthermore, the extension of the manuscript with XAS results (both experimental and theoretical) provides further evidence for the proposed reaction mechanism. The updated PDOS with the Hubbard U term applied to Co's d-states in Figure 4 provides a much clearer picture than the previous version and is additionally in agreement with the experimental and theoretical XAS analysis clarifying the role of symmetry, crystallinity and amorphization for the system.

Overall, the presented manuscript carries a detailed presentation of the amorphization-induced performance increase for Co containing nanosheets with exhaustive support from various experimental and theoretical investigations. In combination with the relatively high-performance compared to other Co-based materials, an acceptance for publication in Nature Comm. can be justified in my opinion. However, before acceptance a few points remain to be clarified in the updated manuscript:

Response: We sincerely thank the reviewer for the positive comments on our work and we are also pleased to clarify the raised issues by the reviewer.

Technical comments:

There are several cases for which the 2p XAS signal of Co²⁺ systems shows temperature dependence.¹⁻³ The temperature dependence is given by low-lying multiplet initial states. Hence, it is essential to provide the temperature for the simulation, which simultaneously needs to be consistent with the experimental conditions to achieve consistency.

Response: We thank the reviewer for the valuable suggestions. The temperature for the simulations is 300 K, which is consistent with the temperature for the

experimental conditions (at room temperature). The temperature for the simulations and experimental conditions has been added in “Method” section.

Moreover, the choice of U_{pd} being smaller than U_{dd} in the case of Co^{2+} O_h is somewhat unusual, and perhaps even unphysical. U_{pd} is generally larger than U_{dd} ⁴ because the presence of the core hole in the final state leads to a contraction of the states resulting in stronger Coulomb repulsion (i.e. the monopole term in the Slater Condon integrals) which is usually given by a value being 1-2 eV larger than U_{dd} . Hence, the authors should explain the choice of $U_{pd} < U_{dd}$.

Response: We thank the reviewer for the valuable comments. Yes, multiplet calculations spontaneously concluded that U_{pd} is generally larger than U_{dd} , because the average distance between two d electrons is usually larger than the distance between a d and a core electron. However, it is still difficult and controversial to accurately obtain the values of U_{dd} and U_{pd} and compare them only by multiplet calculations. Following the reviewer’s suggestion, we used $U_{pd} > U_{dd}$ for the simulations of Co^{2+} O_h symmetry (Figure R1b and Figure 2f in the revised manuscript). The simulation results are still consistent with the proposed reaction mechanism of our work, that is ‘...the symmetry of ligand field around Co atoms partially convert from the octahedral (O_h) to the tetrahedral (T_d) configurations’. Moreover, after simulations, the ratio of T_d and O_h in a-CoO NSs changes from 1 : 4 to 1 : 2.3. The increased ratio of T_d indicates that the impact of amorphization induced alteration of Co d states is more significant and consistent with our proposed mechanism.

Figure R1. Simulations of Co L-edge XANES spectra for the a-CoO NSs and c-CoO NSs, where the simulation of O_h with parameters of (a) the previous version of $U_{dd} = 6$ eV, $U_{pd} = 5$ eV and (b) $U_{dd} = 4$ eV, $U_{pd} = 5$ eV.

In addition, U_{dd} in the multiplet calculations and the U_{dd} in the first principles part of the manuscript differ almost by a factor of 2. This should also be explained or addressed by the authors.

Response: We thank the reviewer for the valuable comments and suggestions. Following the reviewer's suggestion, the empirical values of U_{dd} for simulations of O_h and T_d in the multiplet calculations were revised to 4 eV (Table R1 and Table 2 in revised SI), which is consistent with that in the first principle part.

Site symmetry	Crystal field strength (10Dq)	Charge transfer energy (Δ)	Hopping e_g electrons (t_e)	Hopping t_{2g} electrons (t_t)	Hubbard U value (U_{dd})	Core hole potential (U_{pd})
$Co^{2+} O_h$	1.0	1.0	2.0	1.0	4.0	5.0
$Co^{2+} T_d$	-0.5	6.0	1.0	2.0	4.0	5.5

Table R1. The adjusted parameters for the simulation of Co L-edge XANES spectra by CTM4XAS (Units: eV).

Minor comments

- The phrase 'all calculations' in the 'Theoretical method' part does not capture the complexity the authors put into this manuscript as DFT calculations are only a part of it.

Response: We thank the reviewer for the valuable advices. We have replaced 'all calculations' with 'first principle calculations'.

- There are a few typos in the 'Theoretical method', e.g., 'a pro-', 'Hubbard models'.

Response: We thank the reviewer for the valuable advices. We have replaced 'a pro-' with 'an interface' and changed 'Hubbard models' to 'Hubbard model'.

- What do the authors mean by 'From the PDOS in Fig. 4d, only few t_{2g} electrons under Fermi level lost after Li_2S_4 adsorption.'?

Response: We thank the reviewer for the valuable comments. We tried to compare the difference for number of electrons lost between c-CoO and a-CoO after Li_2S_4

adsorption. The electrons occupying t_{2g} orbitals of Co_{CR} are more stable and not easy lost, because they are at lower energy level (under Fermi level). Therefore, few electrons at the t_{2g} orbitals would lose after Li_2S_4 adsorption, which can be demonstrated by PDOS in Fig. 4d. In contrast, after re-distribution of d -orbitals in a-CoO, the energy level of t_{2g} orbitals is upshifted, and these electrons will be easy lost. We have rewritten the phrase in the revised manuscript as “The PDOS in Fig. 4d indicate that only few t_{2g} (d_{xy} , d_{xz} , d_{yz}) electrons of Co_{CR} lost after Li_2S_4 adsorption, because t_{2g} electrons located at lower energy level (under Fermi level) are stable”

- Unit are missing in Table 2 in SI, as well as details on the applied convolution of transition intensities and screening parameters.

Response: We thank the reviewer for the valuable suggestions. The units (eV) were added in Table 2 in the revised SI.

Details on the applied convolution of transition intensities and screening parameters were added in the notes of Table 2-3 in revised SI. Lorentzian broadening of 0.2 eV half-width half-maximum was applied in each calculated transition intensity to simulate the lifetime broadening of the core hole. And then the curve was convolved with a Gaussian function to simulate the energy resolution of the beamline and other mechanisms of spectral line broadening. The values of c in the Gaussian function were obtained by fitting, together with the ratio of O_h and T_d symmetries (Table R2). The Slater integrals including F_{dd} , F_{pd} and G_{pd} are all set to 1.0 as default. The core (2p) spin-orbit coupling and the valence (3d) spin-orbit coupling are both set to 1.0 as default.

Sample	Symmetry	c
For O_h with $U_{dd} = 6 \text{ eV}$, $U_{pd} = 5 \text{ eV}$		
a-CoO NSs	O_h	0.88
	T_d	0.57
c-CoO NSs	O_h	0.91
For O_h with $U_{dd} = 4 \text{ eV}$, $U_{pd} = 5 \text{ eV}$		
a-CoO NSs	O_h	0.95
	T_d	0.60
c-CoO NSs	O_h	0.95

Table R2. Gaussian function value c for various simulations.

References

(1) Csiszar, S. I.; Haverkort, M. W.; Hu, Z.; Tanaka, A.; Hsieh, H. H.; Lin, H.-J.; Chen, C. T.; Hibma, T.; Tjeng, L. H. Controlling Orbital Moment and Spin Orientation in CoO Layers by Strain. *Phys. Rev. Lett.* 2005, 95 (18), 187205. <https://doi.org/10.1103/PhysRevLett.95.187205>.

- (2) Tanaka, A.; Jo, T. Temperature Dependence of 2 p -Core X-Ray Absorption Spectra in 3 d Transition-Metal Compounds. *J. Phys. Soc. Japan* 1992, 61 (6), 2040-2047. <https://doi.org/10.1143/JPSJ.61.2040>.
- (3) de Groot, F. M. F. X-Ray Absorption and Dichroism of Transition Metals and Their Compounds. *J. Electron Spectros. Relat. Phenomena* 1994, 67 (4), 529-622. [https://doi.org/10.1016/0368-2048\(93\)02041-J](https://doi.org/10.1016/0368-2048(93)02041-J).
- (4) Groot, F. De. Multiplet Effects in X-Ray Spectroscopy. *Coord. Chem. Rev.* 2005, 249 (1-2), 31-63. <https://doi.org/10.1016/j.ccr.2004.03.018>.

Response: We thank the reviewer for providing the important references, which show great help for improving the quality of our multiplet calculation results. We have cited reference (1) and (4) in the revised manuscript.

Reviewer #2 (Remarks to the Author):

All comments were sufficiently addressed by the authors and I can recommend the final manuscript for publication.

Point-by-point Response to the Referees' Comments

Reviewer #2 (Remarks to the Author):

All comments were sufficiently addressed by the authors and I can recommend the final manuscript for publication.

Response: We thank the reviewer very much for the recommendation for publication in *Nature Communications*.